# Magnetoelectric coupling of domains, domain walls and vortices in a multiferroic with independent magnetic and electric order

Marcela Giraldo[1,2], Quintin N. Meier [1,2], Amadé Bortis [1], Dominik Nowak[1], Nicola A. Spaldin [1], Manfred Fiebig [1], Mads C. Weber [1,2] & Thomas Lottermoser [1,2 ✉]

Magnetically induced ferroelectrics exhibit rigidly coupled magnetic and electric order. The ordering temperatures and spontaneous polarization of these multiferroics are notoriously low, however. Both properties can be much larger if magnetic and ferroelectric order occur independently, but the cost of this independence is that pronounced magnetoelectric interaction is no longer obvious. Using spatially resolved images of domains and density-functional theory, we show that in multiferroics with separately emerging magnetic and ferroelectric order, the microscopic magnetoelectric coupling can be intrinsically strong even though the macroscopic leading-order magnetoelectric effect is forbidden by symmetry. We show, taking hexagonal $ErMnO_3$ as an example, that a strong bulk coupling between the ferroelectric and antiferromagnetic order is realized because the structural distortions that lead to the ferroelectric polarization also break the balance of the competing superexchange contributions. We observe the manifestation of this coupling in uncommon types of topological defects like magnetoelectric domain walls and vortex-like singularities.

[1] Department of Materials, ETH Zurich, Zurich, Switzerland. [2] These authors contributed equally: Marcela Giraldo, Quintin N. Meier, Mads C. Weber, Thomas Lottermoser. ✉email: thomas.lottermoser@mat.ethz.ch

The different interdependence of magnetic and electric order in multiferroics with jointly ('type-II') and separately ('type-I') emerging magnetic and ferroelectric order[1] manifests prominently on the level of domains. In the former case, magnetic and electric domains are one-to-one correlated, and every magnetic domain wall is a ferroelectric domain wall, too. This leads to potentially useful phenomena like control of magnetic domains by electric fields, and vice versa[2]. In contrast, in the latter case, the magnetic and ferroelectric domains and domain walls are not intrinsically linked to each other because of the independence of the associated order parameters. Magneto-electric coupling phenomena may still be present, but they are not mandatory. The type-I multiferroic systems of hexagonal ferrites and manganites are model cases for such magnetoelectric coupling phenomena. In ferrites, it was shown that the leading-order, that is, linear magnetoelectric effect dominates the magnetoelectric coupling[3,4]. However, in the isostructural manganites, disregarding any magnetic-field-induced phases and rare-earth ordering[5], the linear magnetoelectric effect is symmetry-forbidden[6]. The demonstration of strong magnetoelectric coupling phenomena in hexagonal manganites, despite this absence of the linear magnetoelectric effect, is at the heart of our work.

We show in experiment and theory that the superexchange interaction drives a pronounced microscopic magnetoelectric interaction. The associated bulk coupling phenomenon is distinctly different from the coupling between[7] or within[8] domain walls proposed earlier. It has significant consequences for the magnetoelectric domain morphology. We furthermore identify three types of magnetic domain walls with different types of magnetic pseudo-vortices at their meeting points. We thus see that the independence of magnetic and electric order in type-I multiferroics can lead to properties that are not open to type-II multiferroics and can thus be beneficial rather than detrimental to their magnetoelectric functionality.

## Results and discussion

Hexagonal ErMnO₃ as representative of the $RMnO_3$ family with $R = $ Sc, Y, In, Dy–Lu is formed by layers of corner-sharing MnO₅ bipyramids alternating with sheets of $Er^{3+}$ ions. A phase transition from the non-polar $P6_3/mmc$ to the polar $P6_3cm$ space group occurs at $T_C \approx 1430\,K$[9]. It results from the collective tilting of every three neighbouring MnO₅ trigonal bipyramids whose apical $O^{2-}$ ions jointly move along the radial direction relative to their central $Er^{3+}$ site. The associated lattice mode has $K_3$ symmetry

and is parameterized by an amplitude $Q$ and azimuthal rotation angle $\Phi$, see Fig. 1. The distortion activates a polar displacement of the $Er^{3+}$ ions along the $c$ axis, giving rise to improper ferroelectricity with a polarization $\mathcal{P} \propto \cos 3\Phi$. This leads to six possible domain states with $\Phi = n \times 60°$ ($n = 0, 1, ..5$), displaying $\Delta\Phi = 60°$ and, hence, alternating polarization between neighbouring domains[10,11]. The system is famous for the formation of ferroelectric pseudo-vortices[12–14], which arise due to the breaking of the effectively continuous symmetry of $\Phi$ close to the Curie temperature[15–17].

At $T_N = 77\,K$, the antiferromagnetic ordering of the $Mn^{3+}$ spins occurs[18] according to the magnetic $K_2$ representation of the non-polar $P6_3/mmc$ structure[19]. The spin angle between nearest $Mn^{3+}$ neighbours in the basal $ab$ planes is 120° with an arrangement of neighbouring planes along $c$ as depicted in Fig. 2. We describe the local direction of the $Mn^{3+}$ spins with the angle $\Psi$, where a change of $\Psi$ corresponds to an in-phase rotation of all the spins in the $ab$ plane, but with an opposite sense of rotation in the upper and lower half of the unit cell[19]. For $Q = 0$, we confirm, using density functional theory (DFT), that the energy of the magnetic order in Fig. 2b is independent of $\Psi$, as required by symmetry. For $Q \neq 0$, however, coupling of the magnetic mode to the structural mode described by a free-energy contribution

$$F_\Psi \propto Q^2 \cos^2(\Psi - \Phi) \qquad (1)$$

is permitted, where the index $\Psi$ emphasizes that the magnetic order adapts to the already established trimerized-polar order. The reduction in symmetry changes the energy landscape from a continuum of equally low-energy states for $\Psi - \Phi$ to only two possible minimum-energy solutions, $\Psi - \Phi = \pm 90°$.

This constraint may appear surprising on first glance since the centrosymmetry of the $Mn^{3+}$ sublattice is not affected by the distortive-ferroelectric transition[10]. Microscopically, however, a correlation between the spin and the polar lattice is established because the magnetic exchange between the $Mn^{3+}$ ions, which determines the antiferromagnetic order, is mediated by the $O^{2-}$ ions[20,21] whose displacement from their original high-symmetry positions is part of the of the structural distortion that leads to the ferroelectric order. Hence, the superexchange projects the polar order into the magnetic system[3], thus establishing a bulk magnetoelectric coupling. It breaks the continuous symmetry in ($\Psi - \Phi$) and centres the high-symmetry point of the magnetic lattice to that of the trimerized-polar lattice. The presence of this coupling is all the more striking as the linear magnetoelectric effect

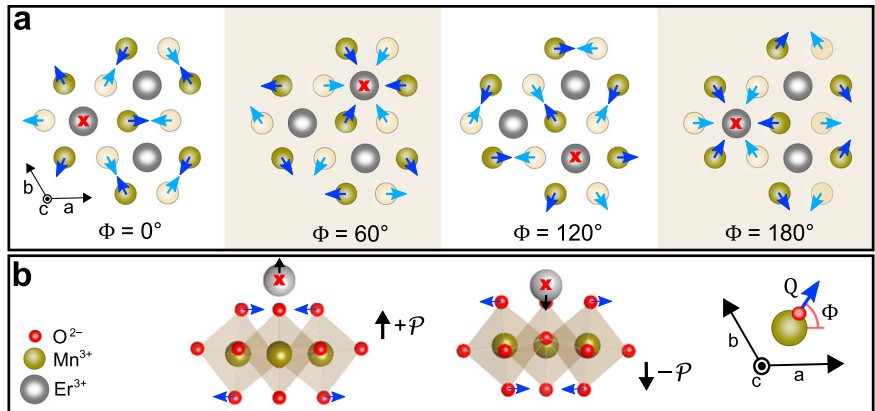

**Fig. 1 Distortive-ferroelectric order of ErMnO₃. a** Top view of the ErMnO₃ unit cell for different values of the azimuthal angle $\Phi$ associated with the tilt of the MnO₅ bipyramids. Dark and bright circles denote $Mn^{3+}$ ions in the upper and lower halves of the unit cell. Dark and bright blue arrows indicate the associated shift of apical $O^{2-}$ ions caused by the bipyramidal tilt. The trimerization centre is marked with a red cross. **b** Coordinated tilting of three corner-sharing MnO₅ bipyramids displaces the $Er^{3+}$ ions along the $c$ axis, and induces improper ferroelectricity ($\pm \mathcal{P}$ polarization states). The inset with the coordinate system defining $Q$ and $\Phi$, in the bottom right, takes the $Mn^{3+}$ ion in the centre of the three $Er^{3+}$ ions in (**a**) as origin.

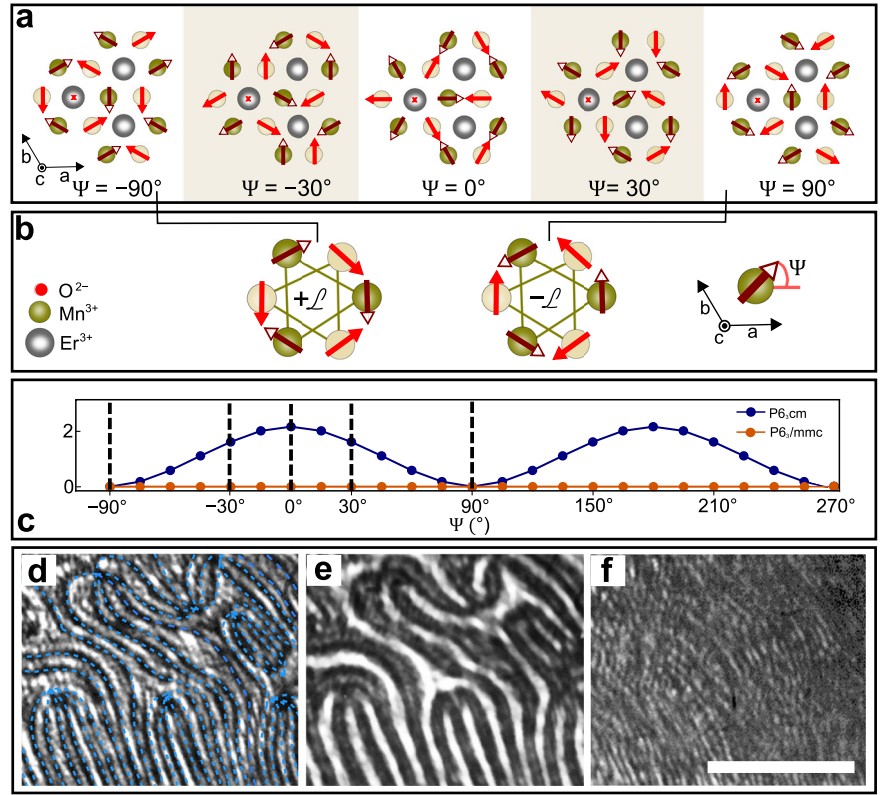

**Fig. 2 Antiferromagnetic order of ErMnO₃. a** Top view of the ErMnO$_3$ unit cell for different values of the Mn$^{3+}$ spin angle $\Psi$. Dark (bright) circles and arrows refer to Mn$^{3+}$ ions in the upper (lower) half of the unit cell. **b** Phenomenological association of the two antiferromagnetic domain states for $\Psi = \pm 90°$ to $+\mathcal{L}$ and $-\mathcal{L}$. The inset with the coordinate system defining $\Psi$, in the bottom right, takes the Mn$^{3+}$ ion in the centre of the three Er$^{3+}$ ions in **a** as origin. **c** Magnetoelectric bulk coupling energy $\Delta E$ as function of the Mn$^{3+}$ spin angle $\Psi$. Calculated using DFT in the absence (orange) and presence (blue) of the distortive-ferroelectric order. **d–f** Spatially resolved distribution of SHG intensity on the same region of a $c$-oriented ErMnO$_3$ sample for the SHG coupling to $\mathcal{P}$, $\mathcal{L}$ and $\mathcal{PL}$, respectively. Black lines—highlighted with dashed blue lines—in **d** indicate $\pm \mathcal{P}$ domain walls, and dark and bright regions in **e** distinguish $\pm \mathcal{L}$ domains. In either case, the brightness difference results from SHG interference processes[24,26]. For the assignment of $\pm \mathcal{P}$ domains in **d**, see Fig. S2. SHG images were taken at room temperature ($\mathcal{P}$) or 20 K ($\mathcal{L}$, $\mathcal{PL}$). Scale bar in **f** is 25 μm.

describing the emergence of a magnetization (electric polarization) proportional to an applied electric (magnetic) field is symmetry-forbidden in ErMnO$_3$[6]. In this aspect, the hexagonal manganites are strinkingly different from the hexagonal ferrites. In LuFeO$_3$, the linear magnetoelectric effect is allowed and involves contributions from a magnetization of the Fe system along the hexagonal axis[3,4], which is absent in ErMnO$_3$.

Symmetry analysis and DFT show that multiferroic structures with $\Psi - \Phi = +90°$ or $-90°$ in Fig. 2a have the same energy; these correspond to two antiferromagnetic structures distinguished by a relative reversal of all Mn$^{3+}$ spins as sketched in Fig. 2b. To see how the degeneracy of the two structures in Fig. 2b influences the domain formation, we experimentally investigate the spatial distribution of ferroelectric and antiferromagnetic domains in ErMnO$_3$. To date, there has been a single such investigation in the system of hexagonal manganites. Zero bulk magnetoelectric coupling was assumed[7,22,23], and the peculiar vortex-like arrangement of the six associated domain states was not considered and could not be spatially resolved[12–14].

We probe the spatial distribution of the domains by optical second-harmonic generation (SHG), yet with ten times higher optical resolution than in the earlier experiments (see 'Methods'). SHG denotes the doubling of the frequency of a light wave in a material. It is a highly symmetry-sensitive process and can therefore distinguish between the different types of long-range order and domain states[24,25]. ErMnO$_3$ features two types of SHG

contributions[7], a ferroelectric one proportional to the electric polarization $\mathcal{P} \propto \cos 3\Phi$ and a multiferroic one proportional to the product $\mathcal{PL}$, where $\mathcal{L} \propto \sin 3\Psi$ is chosen such that it phenomenologically associates the two antiferromagnetic domain states depicted in Fig. 2b to opposite signs. As detailed in 'Methods' and the Supplementary Material, interference of the contributions $\propto \mathcal{P}$ and $\propto \mathcal{PL}$ leads to an antiferromagnetic net SHG contribution $\propto \mathcal{L}$. This approach[24] allows us to distinguish domains with opposite signs of $\mathcal{P}$, $\mathcal{L}$ or $\mathcal{PL}$ as regions of different brightness, or the domain walls themselves as dark channels[26].

In Fig. 2c–e, we see spatially resolved SHG images showing the distribution of the domains for $\mathcal{P}$, $\mathcal{L}$ and $\mathcal{PL}$ in the same area of a $c$-oriented ErMnO$_3$ sample (see 'Methods'). First, the SHG light confirms the relation $\Psi - \Phi = \pm 90°$ derived from Eq. (1) because its polarization reproduces the associated magnetic symmetry[6,7]. The domain configuration in $\mathcal{P}$ yields the familiar distribution of six domains of alternating polarization (see Fig. S2) arranged around a point in which all these domains meet. Strikingly, the domain configuration in $\mathcal{L}$ reproduces this distribution, showing an arrangement of six domains of alternating orientation of the antiferromagnetic order parameter around a vortex-like meeting point. The distribution of $\mathcal{PL}$ is even more striking as it reveals an area of approximately homogeneous brightness without any domain walls, pointing to a $\mathcal{PL}$ single-domain state. The system preserves the value of $(\Psi - \Phi)$ through simultaneous changes of $\Phi$ and $\Psi$ by the same value of $\pm 60°$ when crossing a $\mathcal{P}$ or

$\mathcal{L}$ domain wall. In this sense, $\mathcal{PL}$ is associated to a configuration of hyperdomains at least an order of magnitude larger (see Fig. S3) than the $\mathcal{P}$ and $\mathcal{L}$ domains and not showing their topological-vortex-like distribution.

The preservation of $(\Psi - \Phi)$ immediately disproves the model of a piezomagnetic coupling between strained ferroelectric and locally magnetized antiferromagnetic domain walls proposed earlier[7,22], as the latter would promote the largest possible wall magnetization, which would be accomplished by a change of $\Psi$ by 180° across the wall.

The preservation of $(\Psi - \Phi)$ across the domain walls can be captured in the free energy by adding a gradient term describing the magnetic domain wall energy to Eq. (1) according to

$$F_\Psi = s(\nabla\Psi)^2 + AQ^2\cos^2(\Psi - \Phi), \qquad (2)$$

where $s$, $A > 0$ are coefficients that we calculate from first principles (see 'Methods'). As stated earlier, the second term is minimized by having $\Psi - \Phi = \pm 90°$ in each domain. The energy cost from the first term increases with the total rotation angle $\Delta\Psi$ across the domain wall.

Regarding the change of $\Psi$ across the domain wall we have the following possibilities. (i) $\Delta\Psi = \pm 60°$ if $\Psi$ tracks $\Delta\Phi = \pm 60°$ across the wall; in this case both $\mathcal{P}$ and $\mathcal{L}$ change sign and $\mathcal{PL}$ is preserved. (ii) $\Delta\Psi = \mp 120°$ if only $\mathcal{P}$ (and with it $\mathcal{PL}$) changes sign while $\mathcal{L}$ is preserved. The $\mathcal{PL}$-preserving domain wall has the lower energy cost, consistent with our observations. (iii) As a third possibility, we may have a purely magnetic domain wall within a single ferroelectric domain in which only $\mathcal{L}$ (and with it $\mathcal{PL}$) changes while $\mathcal{P}$ is preserved. This requires $\Delta\Psi = 180°$ to satisfy $\Psi - \Phi = \pm 90°$ and is therefore also unfavourable.

We now investigate the occurrence of the three possible types of magnetic domain walls with $\Delta\Psi = \pm 60°$, $\mp 120°$ or 180° in the same region of the sample after consecutive annealing cycles above $T_N$. The microscopy images in Fig. 3a–d show the ferroelectric $\pm \mathcal{P}$ domains and the antiferromagnetic $\pm \mathcal{L}$ domains along with the analysis in terms of the distribution of $\Phi$ and $\Psi$ in Fig. 3e–h. Note that the direct tracking of the orientation of $\Phi$ and $\Psi$ across a section through the domain wall is prohibited by the optical resolution limit of 1 μm.

The energetically preferred $\Delta\Psi = \pm 60°$ walls are most common and always appear at the same location because of their clamping to the $\Delta\Phi = \pm 60°$ walls, which are not affected by the heating above $T_N$. In Fig. 3b, f, these are the only domain walls present. In addition, Fig. 3c, g reveals a single $\Delta\Psi = \mp 120°$ wall. Even though $\mathcal{L}$ does not change across this wall, it nevertheless represents a magnetic domain wall because $\Psi$ has to change along with $\Phi$ in order to retain $\mathcal{L}$. Finally, Fig. 3d, h shows a single $\Delta\Psi = 180°$ wall across which the brightness of each antiferromagnetic domain is reversed. This is the only unclamped wall, across which only $\mathcal{L}$ and $\Psi$ change, while $\mathcal{P}$ and $\Phi$ do not.

The coexistence of the three types of magnetic walls with the topologically protected network of ferroelectric vortex-like domains leads to additional magnetic pseudo-vortices. According to the analysis in Fig. 3f–h we have three types of these. The domain walls with $\Delta\Psi = \pm 60°$ and $\Delta\Phi = \pm 60°$ coincide and therefore exhibit sixfold magnetic pseudo-vortices in both $\Psi$ and $\mathcal{L}$ (Fig. 3f). The meeting of a $\Delta\Psi = \pm 60°$ wall and a $\Delta\Psi = 180°$ wall can occur as an intersection, which establishes a fourfold magnetic pseudo-vortex in both $\Psi$ and $\mathcal{L}$ (Fig. 3h). Alternatively, the meeting of a $\Delta\Psi = \pm 60°$ wall and a $\Delta\Psi = 180°$ wall can occur as junction with a $\Delta\Psi = \mp 120°$ wall, which establishes a threefold pseudo-vortex in $\Psi$, but not in $\mathcal{L}$ (Fig. 3g).

Finally, to confirm that this interpretation is consistent with the configuration of domains and domain walls seen in Fig. 3, we studied the distribution of the ErMnO$_3$ domains in phase-field

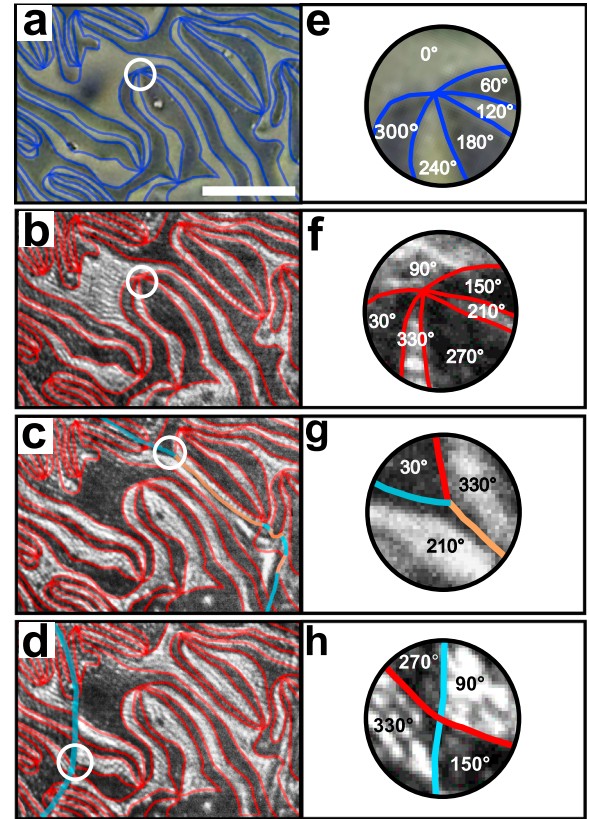

**Fig. 3 Domain walls and pseudo-vortices in ErMnO$_3$. a** Ferroelectric $\pm \mathcal{P}$ domain configuration on a c-oriented ErMnO$_3$ sample by phase-contrast microscopy with domain walls highlighted by blue lines. Scale bar is 25 μm. **b–d** Spatially resolved distribution of $\pm \mathcal{L}$-related SHG on the same region as in **a** after consecutive heating cycles through $T_N$. As in Fig. 2e, dark and bright regions distinguish the $\pm \mathcal{L}$ domains. Red, orange and light blue lines highlight domain walls with $\Delta\Psi = \pm 60°$, $\mp 120°$ and 180°, respectively. The image in **a** is taken at room temperature. Temperature in **b–d** is 20 K. **e–h** Zoom into the ferroelectric and the three types of antiferromagnetic pseudo-vortices discussed in the text with associated values of $\Phi$ in **e** and $\Psi$ in **f–h**.

simulations, based on our model free energy in Eq. (2) and our parameters calculated by DFT. Figure 4 shows the resulting calculated domain configurations in $\Phi$, $\Psi$ and $(\Psi - \Phi)$ and their observable projections $\mathcal{P}$, $\mathcal{L}$ and $\mathcal{PL}$. The results are in excellent agreement with the measured data in Figs. 2 and 3. In particular, all the magnetic pseudo-vortices discussed above are obtained and the $\mathcal{PL}$ hyperdomains are an order of magnitude larger than the domains in $\mathcal{P}$ and $\mathcal{L}$ alone. This confirms that the magnetic domain morphology in ErMnO$_3$ is indeed a direct consequence of the microscopic magnetoelectric bulk coupling.

In conclusion, we see that despite the absence of the linear magnetoelectric effect in the type-I-multiferroic hexagonal manganites, these compounds host a pronounced microscopic bulk magnetoelectric coupling. For our model compound, hexagonal ErMnO$_3$, we show that the interaction between the spins and the lattice occurs because of superexchange, specifically because the coupling between the magnetic Mn$^{3+}$ ions is mediated by the O$^{2-}$ ions, which have different positions in different polar domains. The hidden bulk magnetoelectric coupling in this type-I multiferroic leads to phenomena not open to type-II multiferroics, such as a rich variety in the types and topology of domain walls. Structural distortions determine the magnetic order of many multiferroics. Therefore, a microscopic

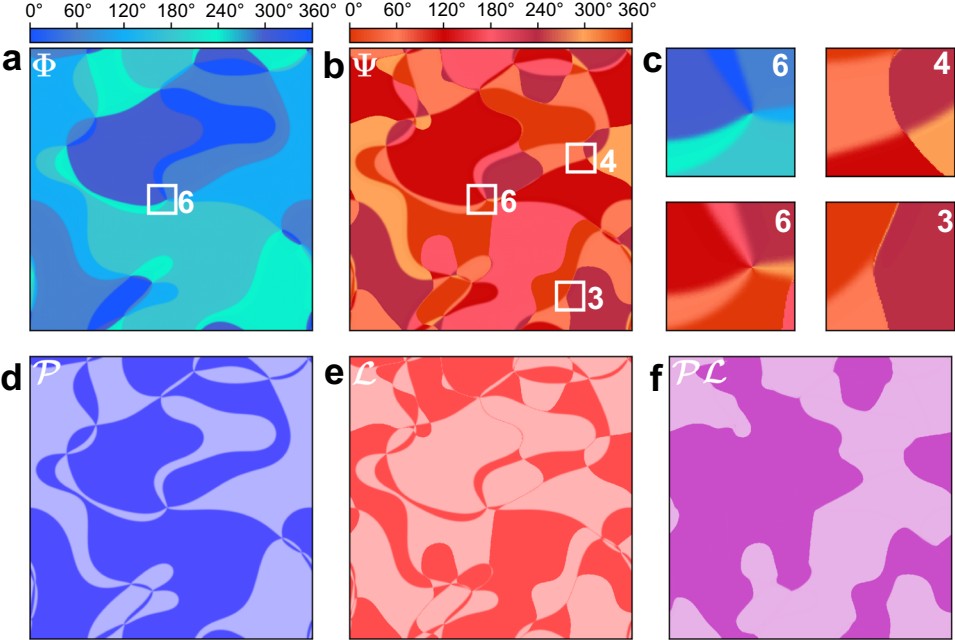

**Fig. 4 Phase-field simulation of the multiferroic domain distribution in ErMnO₃. a** Trimerization domains characterized by the angle Φ. **b** Antiferromagnetic domains characterized by the angle Ψ. **c** Zoom into the different types of pseudo-vortices in **a** and **b** discussed in the text. **d–f** Domain structures in $\mathcal{P}$, $\mathcal{L}$ and $\mathcal{PL}$ resulting from the distribution of Φ and Ψ in **a** and **b**.

| Table 1 Measurement configurations to reveal SHG contributions proportional to $\mathcal{P}$, $\mathcal{PL}$ and $\mathcal{L}$. | | | |
|---|---|---|---|
| **Temperature (T)** | **Sample orientation** | **SHG field** | **Order-parameter sensitivity** |
| $T_N < T < T_C$ | $\mathbf{k} \not\parallel c$ | $\propto \hat{\chi}(\mathcal{P})$ | $\mathcal{P}$ |
| $T < T_N$ | $\mathbf{k} \parallel c$ | $\propto \hat{\chi}(\mathcal{PL})$ | $\mathcal{PL}$ |
| $T < T_N$ | $\mathbf{k} \not\parallel c$ | $\propto (\hat{\chi}(\mathcal{P}) + \hat{\chi}(\mathcal{PL}))$ | $\mathcal{L}$ |

magnetoelectric bulk coupling, demonstrated here for ErMnO₃, should be present in many of these. In particular, we see that the independence of magnetic and electric order in type-I multiferroics is beneficial rather than detrimental to their magnetoelectric functionality because of the greater freedom in coupling the magnetic to the electric domains.

## Methods

**Sample preparation**. We used an ErMnO₃ bulk crystal of about $1 \times 2$-mm² lateral dimensions. The sample was grown by the flux technique and oriented using a Laue x-ray diffractometer. The sample was cut perpendicular to the hexagonal $c$ axis using a diamond saw, thinned down to ~30 μm by lapping with Al₂O₃ powder of 9-μm grain size and chemo-mechanically polished with a colloidal silica slurry until a root-mean-square roughness below 50 nm was reached. Ferroelectric domains with tens of micrometres in size are obtained by heating the crystal across $T_C = 1429$ K and re-cooling it at a rate of about 0.01 K/min[9].

**Optical second-harmonic microscopy**. A transmission SHG setup described in detail elsewhere[24] is used to acquire the spatially resolved SHG images. We use three experimental configurations (see Table 1) to disentangle the different SHG contributions in ErMnO₃[7,27] with **k** denoting the propagation direction of the light.

The SHG signal from $\hat{\chi}(\mathcal{P})$ is usually recorded at room temperature using an angle of 20° between the wavevector and the $c$ axis. Spatial maps reveal the ferroelectric domain configuration. At the domain walls, the opposite sign of the order parameter on either side introduces a relative phase shift of 180° between the respective SHG waves, so that destructive interference distinguishes the domain walls as black lines. The association of opposite $\mathcal{P}$ domains is retrieved from linear phase-contrast microscopy measurements as detailed in Fig. S2. The SHG signal from $\hat{\chi}(\mathcal{PL})$ is recorded below $T_N$. Spatial maps reveal the multiferroic hyperdomain configuration. The SHG signal from $\hat{\chi}(\mathcal{L})$ is obtained from the

interference of the $\mathcal{P}$- and $\mathcal{PL}$-related SHG waves. A phase shift between the two contributions and, thus, a change in the total SHG intensity resulting from the interference of the two contributions, can only occur with a change in $\mathcal{L}$. A change in $\mathcal{P}$ would affect both SHG contributions and therefore retain their relative phase and the resulting SHG intensity. Images were corrected for the variation of intensity across the laser-beam profile.

For the laser excitation we use a Coherent Ellite Duo laser system (1.55-eV fundamental wave, ~120-fs pulse duration, 8-mJ pulse energy, 1-kHz repetition rate). An optical parametric amplifier converts the emission to a photon energy of 1.23 eV and a pulse energy of about 30 μJ. A ×20 long-working-distance microscope objective is used to collect the SHG light emitted from the sample and project it onto a liquid-nitrogen-cooled Jobin Yvon Back Illuminated Deep Depletion CCD camera. Optical filters in the optical path suppress any unwanted frequency components.

**First-principles calculations**. For our first-principles calculations, we use the PBEsol+U approximation of the exchange correlation potential as implemented in the VASP code[28–32]. For the calculation of the magnetic energy landscape, we use $U = 3$ eV. In the Er³⁺ pseudopotential, the $4f$ electrons are treated as core electrons, and therefore rare-earth magnetism is neglected. We use a $k$-point mesh of $6 \times 6 \times 4$ and a plane-wave cutoff energy of 800 eV. All the calculations include spin-orbit coupling. To calculate the energy landscape (Fig. 2), we performed a structural relaxation for each spin configuration with a threshold force of $10^{-4}$ eV/Å² for each atom. We neglect the small out-of-plane tilting that is symmetry-allowed for some of the configurations[19] since preliminary calculations showed that such a tilt only has a very small influence on the total energy. To calculate the free energy introduced in the main text in Eq. (2), the anistropy parameter ($A$) is obtained by performing a least squares fit to calculated spin configurations. The gradient parameter $s = \frac{1}{2}\frac{\partial^2 E_{KS}(k)}{\partial k^2}$ in Eq. (2) is calculated using a long-wavelength expansion of the magnetic order parameter, where $E_{KS}(k)$ is the Kohn–Sham energy for the structure with a magnetic order modulated with a wavevector $k$. Numerically this was done using the spin-spiral implementation in VASP. Normalizing the order parameter Q such that $|Q| = 0$ in the high-symmetry phase and $|Q| = 1$ in the ground state, we obtain the following parameters for Eq. (2): $s = 742$ meVÅ² for the gradient parameter and $A = 2.13$ meV for the anisotropy parameter (per 30 atoms). We note that this is a simplified version of the free energy put forward by Artyukhin et al.[11], including only magnetic configurations of the magnetic $K_2$ representation. We also calculate the free-energy parameters for ErMnO₃ using the full expression[11]:

$$f(\psi_1, \psi_2, \Phi) = \frac{s}{2}\left[(\nabla\psi_1)^2 + (\nabla\psi_2)^2\right] + a\left[\sin^2(\psi_1 - \Phi) + \sin^2(\psi_2 - \Phi)\right] - C_+ \cos(\psi_1 + \psi_2 - 2\Phi) - C_- \cos(\psi_1 - \psi_2),$$
(3)

for which we obtain $s = 742$ meVÅ², $a = 0.027$ meV, $C_+ = -1.05$ meV, $C_- = 0.0023$ meV (per 30 atoms).

**Phase-field modelling**. We use the Landau expansion for the free energy $F(Q, \Phi, \mathcal{P}, \psi_{1,2})$ of the hexagonal manganites introduced by Artyukhin et al.[11] to derive the Ginzburg–Landau (GL) equations[33] for the structural and magnetic order parameters. We use the values given by Artyukhin for the couplings of $Q$, $\Phi$ and $\mathcal{P}$ and values obtained with our DFT simulations for the relation of $\psi_{1,2}$ to $\Phi$ and $\psi_{2,1}$ (see Eq. (3)). Starting from random inital values for all order parameters, we integrate the GL equations applying a semi-implicit Fourier-spectral solver[34], using a grid spacing $h = 0.1$ nm and a time step $dt = 0.1$. We simulate only one layer in the $ab$-plane using a square grid consisting of $n_x \times n_y = 1024 \times 1024$ points. This corresponds to a physical length of the simulated system of about 100 nm. Since the magnetic domain walls have an expected width on this order[11], we have to decrease this value in our simulation to obtain realistic domain patterns. We achieve this by increasing the coupling parameter $a$ in Eq. (3) to $\tilde{a} = 10^3 a$. This reduces the width of magnetic domain walls such that it becomes comparable to the width of the structural walls without changing the topology of the system. Since the ferroelectric phase transition occurs at higher temperature than the magnetic one, we first iterate the trimerization amplitude $Q$ and the phase $\Phi$ as well as the electric polarization $\mathcal{P}$ for $10^4$ steps. We then iterate the spin angles $\psi_{1,2}$ on top of this pattern for $10^4$ steps, resulting in the domain pattern shown in Fig. 4.

**Reporting summary**. Reporting Summary Further Information on research design is available in the Nature Research Reporting Summary linked to this article.

## Data availability
Data that support our findings are available upon request.

## Code availability
The custom code used for the phase-field simulations is available upon request.

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

## Acknowledgements
Equal contributions: M.G., Q.N.M. (experiments and calculations); M.C.W., T.L. (coordination and supervision). The authors thank Dr. Martin Lilienblum for the sample preparation. A.B., M.C.W., M.G. and M.F. received funding by the EU European Research Council (Advanced Grant 694955-INSEETO) and by the Swiss National Science Foundation (Grant No. 200021_178825). Q.N.M. and N.A.S. received funding from ETH Zurich and from the European Research Council (ERC) under the European Union's Horizon 2020 research and innovation programme project HERO under Grant Agreement No. 810451. Computational resources were provided by the Swiss National Supercomputing Center (CSCS) under project numbers s889 and eth3 and by the Euler cluster at ETH Zurich.

## Author contributions
M.G. designed and conducted the experiments. M.G., M.C.W. and T.L. evaluated the experimental data. Q.N.M. and N.A.S. performed the DFT calculations and phenomenological modelling. A.B. developed the phase-field code and performed the phase-field simulations assisted by D.N. The work was conceived by M.F. and T.L and supervised by N.A.S., M.F., M.C.W. and T.L. All authors co-wrote the paper and discussed the results.

## Competing interests
The authors declare no competing interests.
