## [Peer Review File · Nature Communications]

REVIEWER COMMENTS

Reviewer #1 (Remarks to the Author):

In this work, Giraldo et al. experimentally studied the magnetoelectric domain structures in hexagonal multiferroic ErMnO₃. They observed the coupling between antiferromagnetism and ferroelectricity, which was predicted before by theorists. This work pretends to deliver some new physics such as "Superexchange-driven magnetoelectric coupling" in type-I multiferroics. In fact, to my understanding, the story is oversold: the physics revealed in this work is not really new, but had been revealed in previous theoretical works such as Ref. [16] and others. However, the present work indeed provides a very nice experimental verification, which is already important enough for Nat. Commun.. Thus my suggestion is to revise the abstract and introduction a little bit to artless ones, from the Nature style to the Nat. Commun. style.

In addition, I have following technical comments:

1) There are some sentences difficult to understand, which are written in a too fancy style.

For example, "To see how nature handles this ambiguity", does it mean an order parameter with degenerate +/- value?

"It assumed zero bulk magnetoelectric coupling and did not consider the trimerization domain states". I can not capture the physical means.

"Superposition of the contributions P and PL leads to an antiferromagnetic net SHG contribution L.

Does it mean that by reducing the P contribution from PL, then the contribution of L can be derived?

2) There were some works which might be quite related. Please check Nano Lett. 12, 6055 (2012) and Nature Mater. 13, 163 (2014). Please clarify the differences between these works and the current one.

Reviewer #2 (Remarks to the Author):

In this manuscript, the authors discovered a new mechanism for achieving strong magnetoelectric coupling in type-I multiferroics, using a combination of SHG imaging experiments and density-function theory. This reported new mechanism is the magnetic superexchange coupling through the oxygen sites that are involved in the structural distortion for forming the ferroelectric order. They have shown theoretically that the magnetic order through such superexchange coupling has a determined relationship to the polar order and only supports three types of magnetic domain walls. And they have also demonstrated experimentally that there are indeed three types of magnetic domain walls in total in EuMnO₃ and three corresponding magnetic domain vortices, nicely matching the theory. This study represents a very important finding that has been long sought-after by the multiferroics community – strong magnetoelectric coupling at relatively high temperatures. The work is of top high quality, and the manuscript is well-written. Therefore, I would recommend the publication of this work at Nature Communications.

There are a few minor suggestions for the authors to consider.

As the discovered mechanism is unique to type-I multiferroics and is not open to type-II ones, it may be more explicit if the "multiferroics" in the title is updated to "type-I multiferroics".

When defining Φ and Ψ , it may be worth to specify at which MnO cage or Mn site in the unit cell it is defined.

The authors used three experimental geometries/conditions, oblique incidence at $T_N < T < T_C$, normal incidence at $T < T_N$, oblique incidence at $T < T_N$, to couple to three different order parameters, P, PL, and L, respectively. It may be worth devoting a supplementary section to explain why these geometries/conditions pick up the corresponding order parameters, with explicit SH fields expressed in the SH optical susceptibility tensor elements under the proper symmetry point groups.

For the Ψ change (i.e., $\Delta\Psi$) across the three types of magnetic domain walls, are there direct experimental evidence to show their values?

Reviewer #3 (Remarks to the Author):

The article presented by M. Giraldo et al, shows the strong superexchange, present in ErMnO₃ domain walls. The article is interesting, well written and can appeal to the broad audience working in multiferroics or correlated systems. The interpretation and simulations provide a strong basis for future understanding and possible exploitation of these effects in other manganites, thus it can be considered for publication in Nat Comms. Personally, I have very small remarks regarding some minor aspects of the manuscript, which I believe can allow readers to profit even more from the publication. General remarks:

- Please include the SHG spectra of ErMnO₃ in the supplementary section. Marking the signals for the two contributions (Ferroelectric and Multiferroic) and the antiferromagnetic.
- Please provide more numerical details and clear markings of the assumptions, if more convenient to the supplementary section.
- For me, it is not clearly shown what is the magnitude, percentage or estimated value of the magnetoelectric coupling effect at the domain boundaries. Especially when compared with that of ErMnO₃ domains. I understand that the direct relationship between spin and polar ordering shows the statement, but I believe that the "intrinsically strong" aspect, could be explained a bit better in the discussion section.

We thank the Reviewers for the effort invested into reviewing our manuscript, for their positive reception of our work and their constructive comments. In the following, we address point-by-point the input and suggestions for improvement by the reviewers.

Changes to the manuscript are marked in blue for easy tracking.

RESPONSE TO REVIEWER COMMENTS

Reviewer #1:

1.1. In this work, Giraldo et al. experimentally studied the magnetoelectric domain structures in hexagonal multiferroic ErMnO_3 . They observed the coupling between antiferromagnetism and ferroelectricity, which was predicted before by theorists. This work pretends to deliver some new physics such as "Superexchange-driven magnetoelectric coupling" in type-I multiferroics. In fact, to my understanding, the story is oversold: the physics revealed in this work is not really new, but had been revealed in previous theoretical works such as Ref. [16] and others. However, the present work indeed provides a very nice experimental verification, which is already important enough for Nat. Commun.. Thus my suggestion is to revise the abstract and introduction a little bit to artless ones, from the Nature style to the Nat. Commun. style.

Response: Re-reading the manuscript after the months that have passed since submission, we fully agree with the reviewer that the key findings of our work and their relation to Das et al. should be presented more clearly. Indeed, the influence of the trimerization of the magnetic interactions was already discussed in Ref. 16. In doing so, Das et al. focused on those types of magnetic order that exhibit a magnetization along with the electric polarization and permit the leading-order linear magnetoelectric effect.

In contrast, we focus on those systems not exhibiting a magnetization ($M=0$) and strictly forbidding the linear magnetoelectric effect. To some surprise we find that despite this suppression, a pronounced magnetoelectric bulk interaction is maintained by the superexchange interaction. Another difference is that we focus on the consequences of this magnetoelectric bulk coupling, at $M=0$, on the formation of the magnetic domains and domain walls. This leads to novel types of pseudo-vortices and multiferroic hyperdomains. Furthermore, in contrast to Ref. 16, we specifically consider the role of the domain walls in their formation.

We revised the last part of the first paragraph, the introductions on page 2, the discussion of the h-RMnO_3 system on page 4 and the conclusion to bring out the relation and differences to Ref. 16 more clearly now.

In addition, I have following technical comments:

There are some sentences difficult to understand, which are written in a too fancy style.

1.2. For example, "To see how nature handles this ambiguity", does it mean an order parameter with degenerate +/- value?

Response: The system allows for two energetically degenerate antiferromagnetic states, which creates an ambiguous situation. To present this situation more clearly, we changed the wording in lines 82-83 to: *"To see in what form the degeneracy of the two structures in Fig. 2b influences domain formation, we experimentally investigate..."*

1.3. "It assumed zero bulk magnetoelectric coupling and did not consider the trimerization domain states". I cannot capture the physical means.

Response: The sentence refers to former investigations, in which a bulk magnetoelectric coupling was excluded, and the coupling was solely explained as a domain wall effect. Neither did this study consider the structural domain morphology resulting from the trimerization, which the scientific community did not consider as significant at that time. To make this clearer we changed the phrasing in lines 85-87.

1.4. "Superposition of the contributions P and PL leads to an antiferromagnetic net SHG contribution L". Does it mean that by reducing the P contribution from PL, then the contribution of L can be derived?

Response: The SHG signal from L is obtained from the interference of the P- and PL-related SHG waves. A phase shift between the two contributions and, thus, a change in the total SHG intensity resulting from the interference of the two contributions, can only occur with a sign change in L.

A change in P would affect both SHG contributions and therefore retain their relative phase and the resulting SHG intensity. We added a remark in line 95-97 and a paragraph to the Methods section (lines 197-200) and Supplementary information to make this clearer.

1.5. There were some works which might be quite related. Please check Nano Lett. 12, 6055 (2012) and Nature Mater. 13, 163 (2014). Please clarify the differences between these works and the current one.

Response:

The work in Nano Letters discusses a magnetization and linear magnetoelectric effect confined to the domain walls and related to the rare-earth order. In contrast, we only discuss the reorientation of the

magnetic order parameter across a domain wall in the absence of magnetization and rare-earth order. The work on Nature Materials considers a phase where, just as in the work by Das et al. and Du et al., the linear magnetoelectric effect is allowed.

With the revisions in response to point 1.1 we also highlight the difference to the two papers mentioned by the reviewer.

Reviewer #2:

2.1. In this manuscript, the authors discovered a new mechanism for achieving strong magnetoelectric coupling in type-I multiferroics, using a combination of SHG imaging experiments and density-function theory. This reported new mechanism is the magnetic superexchange coupling through the oxygen sites that are involved in the structural distortion for forming the ferroelectric order. They have shown theoretically that the magnetic order through such superexchange coupling has a determined relationship to the polar order and only supports three types of magnetic domain walls. And they have also demonstrated experimentally that there are indeed three types of magnetic domain walls in total in ErMnO₃ and three corresponding magnetic domain vortices, nicely matching the theory. This study represents a very important finding that has been long sought-after by the multiferroics community strong magnetoelectric coupling at relatively high temperatures. The work is of top high quality, and the manuscript is well-written. Therefore, I would recommend the publication of this work at Nature Communications.

Response: We thank the reviewer for the positive and insightful assessment of our work.

2.2. There are a few minor suggestions for the authors to consider.

As the discovered mechanism is unique to type-I multiferroics and is not open to type-II ones, it may be more explicit if the multiferroics in the title is updated to type-I multiferroics.

Response: We had quite some discussion on this point when we prepared the original submission. In the end, we opted against the use of "type-I multiferroic" in the title because general readers who may not be familiar with the type-I/II classification would not understand it. On the other hand, "independent magnetic and electric order" would be clear to general readers. Instead, we introduce the notion of type I/II right at the beginning of the introduction.

2.3. When defining and, it may be worth to specify at which MnO cage or Mn site in the unit cell it is defined.

Response: We added a clarification about the crystallographic sites to the caption of Figures 1 and 2.

2.4. The authors used three experimental geometries/conditions, oblique incidence at $T_N < T < T_C$, normal incidence at $T < T_N$, oblique incidence at $T < T_N$, to couple to three different order parameters, P, PL, and L, respectively. It may be worth devoting a supplementary section to explain why these geometries/conditions pick up the corresponding order parameters, with explicit SH fields expressed in the SH optical susceptibility tensor elements under the proper symmetry point groups.

Response: To facilitate the understanding of the reader, we have added a section about second harmonic generation processes in hexagonal ErMnO_3 to the Supplementary Information.

2.5. For the Ψ change (i.e., $\Delta\Psi$) across the three types of magnetic domain walls, are there direct experimental evidence to show their values?

Response: With our technique, we cannot access the absolute spin angle Ψ , but we provide direct experimental evidence for $\Delta\Psi$ via our SHG interference technique. Since we can identify the sign changes of P, L and PL, individually – which are directly linked to Φ , Ψ and $(\Psi - \Phi)$ (as discussed in the manuscript) – we can use this information to access $\Delta\Psi$ unambiguously.

Reviewer #3:

3.1. The article presented by M. Giraldo et al, shows the strong superexchange, present in ErMnO_3 domain walls. The article is interesting, well written and can appeal to the broad audience working in multiferroics or correlated systems. The interpretation and simulations provide a strong basis for future understanding and possible exploitation of these effects in other manganites, thus it can be considered for publication in Nat Comms. Personally, I have very small remarks regarding some minor aspects of the manuscript, which I believe can allow readers to profit even more from the publication.

Response: We thank the reviewer for the positive and constructive assessment of our work.

General remarks:

3.2. Please include the SHG spectra of ErMnO₃ in the supplementary section. Marking the signals for the two contributions (Ferroelectric and Multiferroic) and the antiferromagnetic.

Response: We have added the SHG response as a function of photon energy for the P- and the PL-related SHG contributions as Supplementary Figure S1, where we also show and discuss their relative intensities.

3.3. Please provide more numerical details and clear markings of the assumptions, if more convenient to the supplementary section.

Response: We have expanded the information on the first-principles calculation as well as the phase-field simulations in the Methods section.

3.4. For me, it is not clearly shown what is the magnitude, percentage, or estimated value of the magnetoelectric coupling effect at the domain boundaries. Especially when compared with that of ErMnO₃ domains. I understand that the direct relationship between spin and polar ordering shows the statement, but I believe that the intrinsically strong aspect, could be explained a bit better in the discussion section.

The bulk magnetoelectric coupling reported in this work is equally present in the domains and the domain walls, there is no additional contribution at the domain walls. We can parametrize the magnetoelectric coupling with the anisotropy parameter A in Eq. 2, where the $A Q^2 \cos^2(\Psi - \Phi)$ term favours values of $\Psi - \Phi = \pm 90$, and thus results in a magnetic domain wall at every structural domain wall where $\Delta \Phi = 60$. From our SHG images in Fig. 3 we know that the size of magnetic domains is much larger than the domain-wall width, so the gradient parameter s does not contribute to the free energy within a domain where $\nabla \Psi = 0$. Because the gradient parameter s in Eq. 2 only sets the width of the magnetic wall and does not affect the preferred value of $\Psi - \Phi$ within a domain, the anisotropy parameter $A = 2.14 \text{ meV}/30\text{atoms}$ alone sets the energy scale of the magnetoelectric coupling.

For clarity, we further elaborate the coupling mechanism in lines 32-35 and 166-167.

REVIEWERS' COMMENTS

Reviewer #1 (Remarks to the Author):

The authors have well addressed my previous comments and done corresponding revisions. Now I can recommend the acceptance.

Reviewer #2 (Remarks to the Author):

The authors have addressed all my questions satisfactorily. Therefore, I recommend the publication of this work at Nature Communications.

Reviewer #3 (Remarks to the Author):

I appreciate the effort and clarifications from the authors regarding my comments. I believe the manuscript is much cleared now and can be accepted in the present form.